# Policy Pre-training for Autonomous Driving via Self-supervised Geometric Modeling

**Penghao Wu**[1,2]**, Li Chen**[1]**, Hongyang Li**[1,3]***Xiaosong Jia**[3,1]**, Junchi Yan**[3,1]**, Yu Qiao**[1]
[1]Shanghai AI Laboratory   [2]University of California at San Diego
[3]MoE Key Lab of Artificial Intelligence, Shanghai Jiao Tong University
{wupenghao, lichen, lihongyang, qiaoyu}@pjlab.org.cn
{jiaxiaosong, yanjunchi}@sjtu.edu.cn
Code:  https://github.com/OpenDriveLab/PPGeo

## Abstract

Witnessing the impressive achievements of pre-training techniques on large-scale data in the field of computer vision and natural language processing, we wonder whether this idea could be adapted in a grab-and-go spirit, and mitigate the sample inefficiency problem for visuomotor driving. Given the highly dynamic and variant nature of the input, the visuomotor driving task inherently lacks view and translation invariance, and the visual input contains massive irrelevant information for decision making, resulting in predominant pre-training approaches from general vision less suitable for the autonomous driving task. To this end, we propose **PPGeo** (Policy Pre-training via Geometric modeling), an intuitive and straightforward fully self-supervised framework curated for the policy pre-training in visuomotor driving. We aim at learning policy representations as a powerful abstraction by modeling 3D geometric scenes on large-scale unlabeled and uncalibrated YouTube driving videos. The proposed PPGeo is performed in two stages to support effective self-supervised training. In the first stage, the geometric modeling framework generates pose and depth predictions simultaneously, with two consecutive frames as input. In the second stage, the visual encoder learns driving policy representation by predicting the future ego-motion and optimizing with the photometric error based on current visual observation only. As such, the pre-trained visual encoder is equipped with rich driving policy related representations and thereby competent for multiple visuomotor driving tasks. As a side product, the pre-trained geometric modeling networks could bring further improvement to the depth and odometry estimation tasks. Extensive experiments covering a wide span of challenging scenarios have demonstrated the superiority of our proposed approach, where improvements range from 2% to even over 100% with very limited data.

## 1 Introduction

Policy learning refers to the learning process of an autonomous agent acquiring the decision-making policy to perform a certain task in a particular environment. Visuomotor policy learning (Mnih et al., 2015; Levine et al., 2016; Hessel et al., 2018; Laskin et al., 2020; Toromanoff et al., 2020) takes as input raw sensor observations and predicts the action, simultaneously cooperating and training the perception and control modules in an end-to-end fashion. For visuomotor policy models, learning tabula rasa is difficult, where it usually requires a prohibitively large corpus of labeled data or environment interactions to achieve satisfactory performance (Espeholt et al., 2018; Wijmans et al., 2019; Yarats et al., 2020).

To mitigate the sample efficiency caveat in visuomotor policy learning, pre-training the visual perception network in advance is a promising solution. Recent studies (Shah & Kumar, 2021; Parisi

---

*Hongyang Li is the correspondence author. This work was in part supported by NSFC (62206172, 62222607), Shanghai Municipal Science and Technology Major Project (2021SHZDZX0102), and Shanghai Committee of Science and Technology (21DZ1100100).

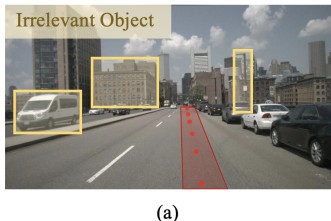 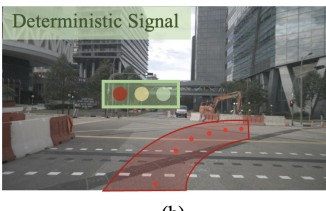 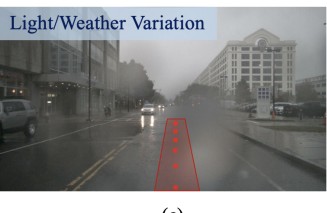

(a)  (b)  (c)

Figure 1: Uniqueness of visuomotor driving policy learning. The planned trajectory is shown as red points. **(a)** static obstacles and background buildings (objects in yellow rectangles) are irrelevant to the driving decision; **(b)** the traffic signal in the visual input (marked with the green box) is extremely difficult to recognize and yet deterministic for control outputs; **(c)** the pre-trained visual encoder has to be robust to different light and weather conditions. Photo credit from (Caesar et al., 2020).

et al., 2022; Xiao et al., 2022; Radosavovic et al., 2022; Shah et al., 2022) have demonstrated that applying popular visual pre-training approaches, including ImageNet (Deng et al., 2009) classification, contrastive learning (He et al., 2020; Chen et al., 2020c), masked image modeling (MIM) (He et al., 2022; Xie et al., 2022; Xue et al., 2022), and language-vision pre-training (Radford et al., 2021), could guarantee superior representation for robotic policy learning tasks, *e.g.*, dexterous manipulation, motor control skills and visual navigation. However, for one crucial and challenging visuomotor task in particular, namely end-to-end autonomous driving[1], the aforementioned predominant pre-training methods may not be the optimal choice (Yamada et al., 2022; Zhang et al., 2022b).

In this paper, we aim to investigate why ever-victorious pre-training approaches for general computer vision tasks and robotic control tasks are prone to *fail* in case of end-to-end autonomous driving. For conventional pre-training methods in general vision tasks, *e.g.*, classification, segmentation and detection, they usually adopt a wide range of data augmentations to achieve translation and view invariance (Zhang et al., 2016; Wu et al., 2018). For robotic control tasks, the input sequence is generally of small resolution; the environment setting is simple and concentrated on objects (Parisi et al., 2022; Radosavovic et al., 2022). We argue that the visuomotor driving investigated in this paper, is sensitive to geometric relationships and usually comprises complex scenarios.

As described in Fig. 1(a), the input data often carry irrelevant information, such as background buildings, far-away moving vehicles, nearby static obstacles, *etc.*, which are deemed as noises for the decision making task. To obtain a good driving policy, we argue that the desirable model should only concentrate on particular parts/patterns of the visual input. That is, taking heed of direct or deterministic relation to the decision making, *e.g.*, traffic signals in Fig. 1(b). However, concurrent pre-training approaches fail to fulfill such a requirement. There comes a natural and necessary demand to formulate a pre-training scheme curated for end-to-end autonomous driving. We attempt to pre-train a visual encoder with a massive amount of driving data crawled freely from the web, such that given limited labeled data, downstream applications could generalize well and quickly adapt to various driving environments as depicted in Fig. 1(c).

The pivotal question is *how to introduce driving-decision awareness into the pre-training process to help the visual encoder concentrate on crucial visual cues for driving policy*. One may resort to directly predicting ego-motion based on single frame sensor input, constraining the network on learning policy-related features. Previous literature tackles the supervision problem with pseudo labeling training on either an open dataset (Zhang et al., 2022b) or the target domain data (Zhang et al., 2022a). However, pseudo labeling approaches suffer from noisy predictions from poorly calibrated models - this is true especially when there exists distinct domain gap such as geographical locations and traffic complexities (Rizve et al., 2020).

To address the bottleneck aforementioned, we propose **PPGeo** (**P**olicy **P**re-training via **Geo**metric modeling), a fully self-supervised driving policy pre-training framework to learn from unlabeled and uncalibrated driving videos. It models the 3D geometric scene by jointly predicting ego-motion, depth, and camera intrinsics. Since directly learning ego-motion based on single frame input along with depth and intrinsics training from scratch is too difficult, it is necessary to separate the visual encoder pre-training from depth and intrinsics learning in two stages. In the first stage, the ego-motion

---

[1]We use *end-to-end autonomous driving* and *visuomotor autonomous driving* interchangeably in this paper.

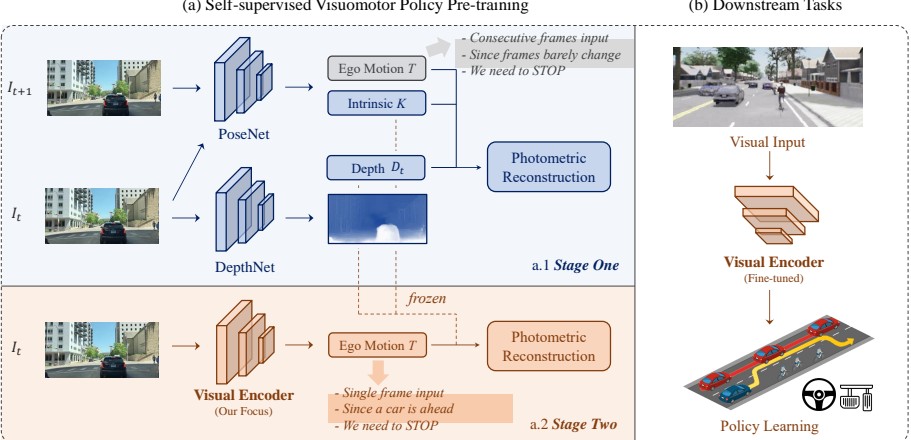

Figure 2: Overview of **PPGeo**. **(a)** We focus on pre-training an effective visual encoder to encode driving policy related information by predicting ego-motion based on single frame input (a.2 Stage Two). As achieving such a goal without labels is non-trivial, the visual encoder is obtained with the aid of a preceding procedure (a.1 Stage One) with temporal inputs and two sub-networks (pose and depth). In this illustrative example, the ego-vehicle needs to take action of STOP. The ego-motion in (a.1) is inferred by judging two consecutive frames barely change; whilst the ego-motion in (a.2) is predicted based on single visual input - focusing on driving policy related information. As such, the visual encoder could be fine-tuned and applied to a wide span of downstream tasks in **(b)**.

is predicted based on consecutive frames as does in conventional depth estimation frameworks (Godard et al., 2017; 2019). In the second stage, the future ego-motion is estimated based on the single frame by a visual encoder, and could be optimized with the depth and camera intrinsics network well-learned in the first stage. As such, the visual encoder is capable of inferring future ego-motion based on current input alone. The pre-trained visual encoder could be well adopted for downstream driving tasks since it captures driving policy related information. As a side product, the depth and pose networks could be utilized as new initial weights for depth and odometry estimation tasks, bringing in an additional performance gain. To sum up, our **key contributions** are three-fold:

- We propose a pre-training paradigm curated for various visuomotor driving tasks. To the best of our knowledge, this is the first attempt to achieve a fully self-supervised framework *without* any need of pseudo-labels[2], leveraging the effect of pre-training by large-scale data to the full extent.

- We devise a visual encoder capable of predicting ego-motion based on single visual input, being able to extract feature representations closely related to driving policy. Such a design of visual encoder is flexible to extend to various downstream applications.

- We demonstrate the superiority of our approach on a set of end-to-end driving scenarios, covering different types and difficulty levels. The performance in terms of various metrics is improved from 2% to even over 100% in challenging cases with very limited data.

## 2 METHODOLOGY

### 2.1 OVERVIEW

The visuomotor policy learning for autonomous driving targets generating a policy $\pi$, such that it makes driving decisions, *e.g.*, control actions or planned trajectory, from visual observation $\mathbf{x}$. Our goal is to pre-train a visual encoder $\phi(\mathbf{x})$, which maps the raw image input to a compact representation containing important information for driving decision making. The representation is then utilized by the policy $\pi(\phi(\mathbf{x}))$ to perform driving tasks. As shown in Fig. 2, our pre-training method pre-trains the visual encoder on unlabeled driving videos via two stages in a self-supervised manner.

---

[2]Pseudo-labels here mean using another model trained on additional labeled data to create "artificial" labels for the unlabeled dataset.

## 2.2 Two-stage Self-supervised Training

**Stage One: Self-supervised Geometric Modeling.** During the first stage, given a target image $I_t$ and source images $I_{t'}$ in a sequence, we jointly estimate the depth of the target image, the intrinsics of the camera, and the 6-DoF ego-motion between these two frames. Given the estimations, we are able to model the 3D geometry of the scene, and reconstruct the target image by projecting pixels in the source images. Formally, the pixel-wise correspondence between $I_t$ and $I_{t'}$ is calculated as:

$$\mathbf{p}_{t'} = \mathbf{K}\mathbf{T}_{t\to t'}\mathbf{D}_t(\mathbf{p}_t)\mathbf{K}^{-1}\mathbf{p}_t, \tag{1}$$

where $\mathbf{p}_t$ and $\mathbf{p}_{t'}$ are the homogeneous coordinates of the pixel in $I_t$ and $I_{t'}$ respectively, $\mathbf{K}$ is the predicted camera intrinsic matrix, and $\mathbf{D}_t(\mathbf{p}_t)$ represents the predicted depth value at pixel $p_i$ in $I_t$. With this relationship, the target image $I_{t'\to t}$ could be reconstructed with pixels in $I_{t'}$, and be optimized by the photometric reconstruction error. Following Godard et al. (2019), we choose two images adjacent to the current frame as the source images, *i.e.*, $t' \in \{t-1, t+1\}$.

The DepthNet consists of a common encoder-decoder structure (Godard et al., 2019) and estimates the depth map of the input image. Two images are stacked together and fed into the encoder of the PoseNet, whose bottleneck feature is then utilized to predict the camera intrinsics and the ego-motion via two separate MLP-based heads. For camera intrinsics estimation, optical center $(c_x, c_y)$ and focal lengths $f_x, f_y$ are regressed similarly as in Gordon et al. (2019); Chanduri et al. (2021).

**Stage Two: Visuomotor Policy Pre-training.** After the first stage of training, the DepthNet and PoseNet are well trained and fitted to the driving video data. Then, in the second stage, we replace the PoseNet for ego-motion estimation with the visual encoder $\phi(\mathbf{x})$ prepared for downstream driving policy learning tasks. Now the visual encoder only takes a single frame image as input and predicts ego-motion between the current frame and subsequent frame.

Specifically, the visual encoder estimates the ego-motion $T_{t\to t+1}$ based on $I_t$ alone and $T_{t\to t-1}$ based on $I_{t-1}$ followed by an inverse operation, respectively. The visual encoder is optimized by the photometric reconstruction error similar to the first stage, aside from a modification where the DepthNet and the intrinsics estimation are frozen and not backpropagated. This is empirically observed towards better performance. By doing so, the visual encoder is enforced to learn the actual driving policy, since the ego-motion between two consecutive frames is straightforwardly related to the driving decision or action taken at the current timestamp.

One might argue that the PoseNet trained in the first stage could provide pseudo motion labels, with which the visual encoder could be directly supervised. However, the ego-motion predicted from the PoseNet is too sparse compared with the geometric projection approach. In our pipeline, every pixel provides supervision for the visual encoder so that inaccurate depth estimation in some pixels could be mitigated by the accurate ones, *i.e.*, it constructs a "global" optimization. In contrast, direct supervision from the PoseNet would be greatly affected by the undesirable prediction inaccuracy and noise results.

Thus far, the backbone of visual encoder $\phi(\mathbf{x})$ has gained knowledge about the driving policy from the diverse driving videos. It can then be applied to downstream visuomotor autonomous driving tasks as the initial weights. Besides, the DepthNet and PoseNet trained on this large corpus of uncalibrated video data could also be utilized in depth and odometry estimation tasks.

## 2.3 Loss Function

Following Godard et al. (2019), the loss function is comprised of the photometric loss and the smoothness loss. The photometric error is comprised of an $\ell_1$ term and an SSIM (structural similarity index measure) term (Wang et al., 2004):

$$\ell_{pe} = \frac{\alpha}{2}(1 - \text{SSIM}(I_t, I_{t'\to t})) + (1-\alpha)\ell_1(I_t, I_{t'\to t}), \tag{2}$$

where we set $\alpha = 0.85$ following the practice (Godard et al., 2017; 2019). The smooth loss is:

$$\ell_s = |\partial_x d_t^*|e^{-|\partial_x I_t|} + |\partial_y d_t^*|e^{-|\partial_y I_t|}, \tag{3}$$

where $d_t^*$ is the mean-normalized inverse depth map. We also adopt the minimum reprojection loss and auto-masking scheme (Godard et al., 2019) to improve self-supervised depth estimation.

## 3 EXPERIMENTS

All pre-training experiments are conducted on the hours-long unlabeled YouTube driving videos (Zhang et al., 2022b). It covers different driving conditions *e.g.*, geographical locations and weather. We sample 0.8 million frames in total at 1 Hz for training. For the first stage in PPGeo pipeline, we train the model for 30 epochs by Adam (Kingma & Ba, 2015) optimizer with a learning rate of $10^{-4}$ which drops to $10^{-5}$ after 25 epochs. For the second stage, the encoder is trained for 20 epochs using the AdamW (Loshchilov & Hutter, 2017) optimizer. A cyclic learning rate scheduler is applied with the learning rate ranging from $10^{-6}$ to $10^{-4}$. The batch size for both stages is 128. We use data augmentations including ColorJitter, RamdomGrayScale, and GaussianBlur.

### 3.1 DESCRIPTION ON COMPARED BASELINES

We use ResNet-34 (He et al., 2016) as the encoder and load different pre-trained weights for the initialization of downstream tasks. We compare PPGeo with pre-training methods including:

**Random.** We use the default Kaiming initialization (He et al., 2015) for convolution layers and constant initialization for batchnorms.

**ImageNet.** We use the model weight provided by Torchvision (Marcel & Rodriguez, 2010), which is pre-trained with the classification task on ImageNet (Deng et al., 2009).

**MIM.** The model is pre-trained with the masked image modeling method on the YouTube driving video, which tries to reconstruct images with random masked-out patches. SimMIM (Xie et al., 2022) is adopted as it is suitable for convolutional networks.

**MoCo.** We pre-train the model using MoCo-v2 (Chen et al., 2020c) on the YouTube driving videos. We exclude RandomResizedCrop and RandomHorizontalFlip augmentations as they are not suitable for the driving task.

**ACO.** Following Zhang et al. (2022b), it is pre-trained using action-conditioned contrastive learning on the YouTube driving videos. ACO trains an inverse dynamic model to generate pseudo steer labels for driving videos, based on which steer-based discrimination is added on top of MoCo-v2.

**SelfD.** SelfD (Zhang et al., 2022a) is not a pre-training method strictly since it needs to train the whole policy model on the driving video for each task, while other pre-training methods aforementioned provide a general pre-training visual model for all tasks. We still include it for comparison due to its close relationship to our target. Specifically, we follow Zhang et al. (2022a) to train the model for each task with the following pipeline: training on the task data → training on the YouTube data with pseudo-label → fine-tuning on the task data.

### 3.2 DESCRIPTION ON DOWNSTREAM AUTONOMOUS DRIVING TASKS

We carry out experiments under (1) three imitation learning based closed-loop driving tasks in CARLA (Dosovitskiy et al., 2017), (2) one reinforcement learning based driving task in CARLA, and (3) an open-loop planning task on real-world autonomous driving dataset nuScenes (Caesar et al., 2020), to fully validate the effectiveness of PPGeo. We briefly describe each task below.

**Navigation.** It corresponds to the goal-conditioned navigation task in the CoRL2017 benchmark (Dosovitskiy et al., 2017). The agent is trained in Town01 and tested in Town02 with unseen weather, and there are no other traffic participants. We use different sizes of training data (from 4K to 40K) following Zhang et al. (2022b) to evaluate the generalization ability of pre-trained visual encoders when labeled data is limited and conduct the closed-loop evaluation. The evaluation metric is success rate, denoting the portion of 50 pre-defined routes finished without any collision. And traffic lights are ignored here. CILRS (Codevilla et al., 2019), a classic image based end-to-end autonomous driving model, is adopted for training and evaluation.

**Navigation Dynamic.** This is the navigation dynamic task in the CoRL2017 benchmark (Dosovitskiy et al., 2017). The setting differentiates from Navigation that there are other dynamic objects such as randomly generated vehicles, which substantially increases the difficulty of driving safety.

**Leaderboard Town05-long.** This challenging and realistic benchmark corresponds to the Leader-Board benchmark (CARLA, 2022). We collect 40K training data in Town01, 03, 04, 06 and evaluate on 10 routes in the unseen Town05 (Prakash et al., 2021). Due to the challenging scenarios in this

Table 1: The Successful Rate of the closed-loop Navigation task (mean by 3 random trials).

| Pre-train Method | Navigation - # of training samples | | | |
|---|---|---|---|---|
| | 10% (4K) | 20% (8K) | 40% (16K) | 100% (40K) |
| Random | $0.0 \pm 0.0$ | $9.6 \pm 5.2$ | $15.3 \pm 4.5$ | $73.3 \pm 2.3$ |
| ImageNet | $24.7 \pm 2.0$ | $42.0 \pm 2.0$ | $69.3 \pm 6.4$ | $87.3 \pm 4.6$ |
| MIM | $4.7 \pm 1.2$ | $8.0 \pm 0.0$ | $31.3 \pm 2.3$ | $57.3 \pm 3.1$ |
| MoCo | $7.7 \pm 2.1$ | $39.3 \pm 9.2$ | $48.7 \pm 4.2$ | $69.3 \pm 1.2$ |
| ACO | $24.0 \pm 2.0$ | $44.0 \pm 1.2$ | $71.3 \pm 1.2$ | $92.0 \pm 3.5$ |
| SelfD | $12.0 \pm 0.0$ | $32.0 \pm 0.0$ | $50.7 \pm 2.3$ | $62.7 \pm 1.2$ |
| **PPGeo** (ours) | $\mathbf{42.0 \pm 2.0}$ | $\mathbf{73.3 \pm 6.1}$ | $\mathbf{91.3 \pm 1.2}$ | $\mathbf{96.7 \pm 1.2}$ |

task, we evaluate different pre-training approaches with the state-of-the-art image-based autonomous driving model TCP (Wu et al., 2022). The metrics of this task are Driving Score, Route Completion, and Infraction Score (all the higher the better), where the main metric Driving Score is the product of Route Completion and Infraction Score.

**Reinforcement Learning.** Proximal Policy Optimization (PPO) (Schulman et al., 2017) is used to train the CILRS (Codevilla et al., 2019) model initialized with different pre-trained weights in CARLA Town01 environment. The reward shaping details follow Roach (Zhang et al., 2021). We also conduct experiments to freeze the pre-trained visual encoder during training to further study the effectiveness of the pre-trained feature representations.

**nuScenes Planning.** This task involves trajectory planning in real-world dataset nuScenes (Caesar et al., 2020). Given the current visual input, the model plans a 3-second trajectory (0.5 Hz), and the planned trajectory is compared with the ground truth log. We also calculate the collision rate, where a collision is defined as overlaps with future vehicles and pedestrians based on planned waypoints. The planning model used here is comprised of a visual encoder and a GRU-based planner to predict each waypoint auto-regressively. We use the official train-val split for training and evaluation.

### 3.3 NUMERIC COMPARISON ON DOWNSTREAM TASKS

For imitation learning based closed-loop driving tasks, the evaluation results are shown in Table 1-3. We present the plot between episode return and environment steps of each method in Fig. 3 for the reinforcement learning experiments. The open-loop nuScenes planning results are provided in Table 4. We could observe that PPGeo outperforms other baselines by a large margin in *all* tasks.

Note that the model is tested under a different number of fine-tuning samples from 10% (4K) to full 40K in the Navigation and Navigation Dynamic tasks. In the case of the particularly small size of training samples, PPGeo still demonstrates competitive performance and has a larger improvement gap of over 100%. This validates the generalization ability of the pre-trained visual encoder, which is important when adapting to a new environment with very limited labeled data. In the more challenging and real-world style Leaderboard Town05-long task in Table 3, the model pre-trained with our method achieves the highest driving score and infraction score. PPGeo well handles cases where the agent needs to stop, leading to much fewer vehicle collisions and red light infractions.

Since ACO considers steering angles only during pre-training, its performance degrades on more challenging scenarios where brake and throttles are also important. SelfD performs slightly better than ACO in complex cases while it significantly degenerates when the task data is limited, as affected by the unsatisfying pseudo labeling model. ImageNet pre-training also shows competitive performance, which might credit to its ability of finding salient objects in the scene when the input contains little irrelevant information (see examples in Sec. 3.5).

### 3.4 DEPTH AND ODOMETRY ESTIMATION

In this part, we explore whether the large-scale training on uncalibrated data could benefit the depth and odometry estimation models as well and validate the effectiveness of first-stage training. Specifically, we employ the DepthNet and PoseNet trained after the first stage as initial weights for Monodepthv2 (Godard et al., 2019), and conduct experiments on KITTI (Geiger et al., 2012). Results in Table 5 indicate that pre-training on large-scale driving videos could bring performance improvement to both depth and odometry estimation tasks, which is an additional harvest of our pre-training framework. We refer readers to Godard et al. (2019) for details about the metrics of these tasks.

Table 2: The Successful Rate of the closed-loop Navigation Dynamic (mean by 3 random trials).

| Pre-train Method | Navigation Dynamic - # of training samples | | | |
|---|---|---|---|---|
| | 10% (4K) | 20% (8K) | 40% (16K) | 100% (40K) |
| Random | $0.0 \pm 0.0$ | $2.0 \pm 0.0$ | $10.0 \pm 0.0$ | $32.0 \pm 8.0$ |
| ImageNet | $10.7 \pm 1.2$ | $28.7 \pm 5.0$ | $64.7 \pm 2.3$ | $72.7 \pm 1.2$ |
| MIM | $7.3 \pm 1.2$ | $10.3 \pm 2.5$ | $14.7 \pm 3.1$ | $58.7 \pm 1.2$ |
| MoCo | $4.7 \pm 1.2$ | $12.0 \pm 4.0$ | $28.0 \pm 5.3$ | $66.7 \pm 2.3$ |
| ACO | $8.0 \pm 1.2$ | $12.0 \pm 0.0$ | $22.0 \pm 2.0$ | $47.3 \pm 5.0$ |
| SelfD | $8.0 \pm 0.0$ | $29.3 \pm 1.2$ | $38.0 \pm 1.6$ | $59.3 \pm 6.4$ |
| **PPGeo** (ours) | $\mathbf{23.3 \pm 1.2}$ | $\mathbf{34.0 \pm 5.3}$ | $\mathbf{71.3 \pm 1.2}$ | $\mathbf{84.0 \pm 5.3}$ |

Table 3: Closed-loop Leaderboard Town05-long task results. Besides three main metrics, infraction details are also reported (all the lower the better). Evaluation repeats 3 times with the mean reported.

| Pre-train Method | **Driving Score** | Infraction Score | Route Completion | Collisions pedestrian | Collisions vehicle | Collisions layout | Off-road violations | Agent blocked | Red light violations |
|---|---|---|---|---|---|---|---|---|---|
| Random | 33.50±1.67 | 0.65±0.02 | 60.49±2.93 | 0.09±0.07 | 1.16±0.40 | 0.00±0.00 | 0.44±0.13 | 0.97±0.09 | 0.53±0.12 |
| ImageNet | 41.29±3.20 | 0.77±0.03 | 57.52±4.87 | 0.00±0.00 | 0.71±0.20 | 0.11±0.15 | 0.15±0.01 | 1.01±0.16 | 0.29±0.10 |
| MIM | 36.39±0.21 | 0.72±0.04 | 61.75±2.26 | 0.14±0.11 | 0.91±0.12 | 0.04±0.07 | 0.18±0.17 | 0.87±0.03 | 0.14±0.11 |
| MoCo | 32.10±2.04 | 0.65±0.02 | 64.09±4.01 | 0.13±0.11 | 0.79±0.16 | 0.00±0.00 | 0.49±0.07 | 0.81±0.15 | 0.45±0.13 |
| ACO | 33.05±3.05 | 0.67±0.06 | 59.52±3.21 | 0.00±0.00 | 0.69±0.28 | 0.05±0.07 | 0.54±0.05 | 0.94±0.08 | 0.73±0.10 |
| SelfD | 38.76±3.02 | 0.65±0.03 | **68.72±7.36** | 0.17±0.07 | 0.84±0.18 | 0.00±0.00 | 0.32±0.03 | 0.75±0.15 | 0.12±0.08 |
| **PPGeo** | **47.44±5.63** | **0.79±0.08** | 65.05±5.11 | 0.04±0.05 | 0.54±0.29 | 0.00±0.00 | 0.16±0.11 | 0.76±0.10 | 0.04±0.05 |

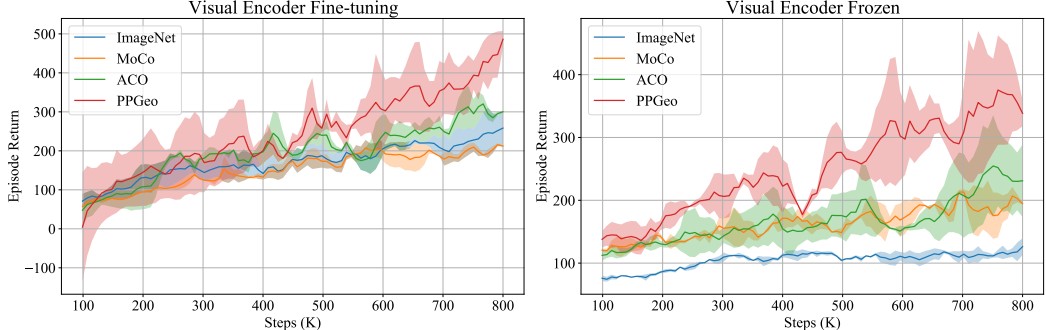

Figure 3: Learning curves of the RL agents using PPGeo and three other best pre-training baselines. Left: the pre-trained visual encoder is jointly fine-tuned during RL training; Right: the visual encoder is frozen during RL training. The episode return is the mean with standard deviation in shade across three runs with different random seeds.

Table 4: Open-loop nuScenes planning results. We evaluate the $\ell_2$ distance between model predictions and the ground truth trajectory and collision rate in horizons from 1 second to 3 seconds.

| Pre-train Method | L2 (m) ↓ | | | Collision Rate (%) ↓ | | |
|---|---|---|---|---|---|---|
| | 1s | 2s | 3s | 1s | 2s | 3s |
| Random | 1.621 | 2.722 | 3.851 | 0.550 | 1.779 | 3.375 |
| ImagNet | 1.331 | 2.202 | 3.086 | 0.315 | 0.550 | 1.366 |
| MIM | 1.412 | 2.357 | 3.331 | 0.297 | 0.622 | 1.507 |
| MoCo | 1.528 | 2.545 | 3.585 | 0.560 | 1.235 | 2.390 |
| ACO | 1.496 | 2.496 | 3.519 | 0.446 | 1.178 | 2.223 |
| SelfD | 1.419 | 2.359 | 3.316 | 0.353 | 0.923 | 2.044 |
| **PPGeo** (ours) | **1.302** | **2.154** | **3.018** | **0.270** | **0.425** | **0.941** |

## 3.5 VISUALIZATION RESULTS

Here we provide heatmaps of the feature representations learned by different pre-training methods using Eigen-Cam (Muhammad & Yeasin, 2020) to show the attended regions in Fig. 4. In many cases (Row 1&2), our model mainly concentrates on the lane in front of the ego vehicle, which is highly related to driving. And our model PPGeo well captures the specific cues causing the brake

Table 5: Improvement from our pre-training method on depth and odometry estimation tasks.

| Pre-train Method | Depth Estimation | | | | | | | Odometry Estimation | |
|---|---|---|---|---|---|---|---|---|---|
| | abs_rel ↓ | sq_rel ↓ | rmse ↓ | rmse_log ↓ | a1 ↑ | a2 ↑ | a3 ↑ | Sequence 09 ↓ | Sequence 10 ↓ |
| ImageNet | 0.118 | 0.902 | 4.873 | 0.196 | 0.871 | 0.958 | 0.981 | 0.017±0.010 | 0.015±0.010 |
| **PPGeo** | **0.114** | **0.805** | **4.599** | **0.186** | **0.874** | **0.962** | **0.984** | **0.016±0.009** | **0.013±0.009** |

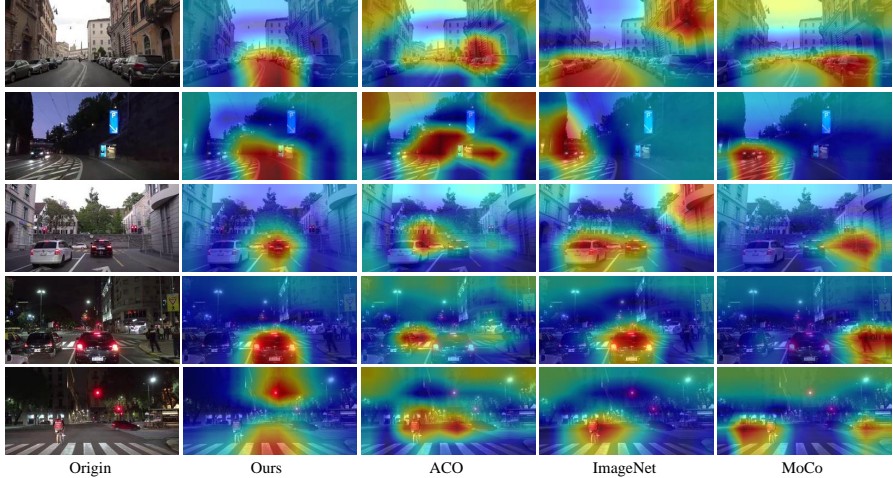

Figure 4: Eigen-Cam (Muhammad & Yeasin, 2020) activation maps of the learned representation from different pre-training methods on the driving video data.

Table 6: Ablative study on key designs of PPGeo on the Navigation task.

| # | Experiment | Navigation - # of training samples | | | |
|---|---|---|---|---|---|
| | | 10% (4K) | 20% (8K) | 40% (16K) | 100% (40K) |
| 1 | Single stage | 24.2 ± 2.0 | 53.3 ± 1.2 | 79.3 ± 4.2 | 92.7 ± 2.3 |
| 2 | No frozen in 2nd stage | 32.7 ± 1.2 | 58.0 ± 2.0 | 86.0 ± 2.1 | 92.0 ± 2.0 |
| 3 | PoseNet direct supervision | 18.0 ± 2.0 | 52.0 ± 2.0 | 76.7 ± 1.2 | 90.0 ± 0.0 |
| 4 | **PPGeo** | **42.0 ± 2.0** | **73.3 ± 6.1** | **91.3 ± 1.2** | **96.7 ± 1.2** |

action including front vehicles (Row 3&4) and traffic lights (Row 5). We also observe that the model pre-trained with ImageNet classification tends to capture salient objects in the image. This is helpful when the salient objects are straightforwardly related to the driving decision (Row 4); but it may focus on wrong objects when the input contains other irrelevant information (Row 2&3).

## 3.6 ABLATIVE STUDY

We conduct ablative study as to different designs of PPGeo on the Navigation task in Table 6. Training the visual encoder and DepthNet in a single stage simultaneously (Row 1) leads to an inferior performance, indicating that it is quite challenging for the visual encoder to learn the correct ego-motion if depth estimation is also trained from scratch. Moreover, jointly optimizing the DepthNet in the second stage (Row 2, not frozen) degrades the depth estimation quality and harms the performance. In Row 3, we observe that utilizing the PoseNet obtained in the first stage to provide pseudo label supervision directly leads to inferior results, since an inaccurate pseudo label impairs the learning process to great extent.

## 4 RELATED WORK

**Pre-training for NLP and General Vision.** Pre-training or representation learning has proved to be an essential key to the success of artificial intelligence. In the field of Natural Language Processing (NLP), with the powerful capability of Transformer (Vaswani et al., 2017), pre-training on large-scale datasets with large models then fine-tuning on downstream tasks has become the dominant paradigm (Kenton & Toutanova, 2019; Brown et al., 2020). As for the field of Computer Vision, training specific downstream tasks with the supervised pre-trained weights of visual encoder via

ImageNet classification task is widely adopted. Recently, unsupervised and self-supervised learning methods such as contrastive learning (He et al., 2020; Chen et al., 2020c;b) and masked image modeling (Bao et al., 2021; He et al., 2022; Xie et al., 2022; Peng et al., 2022; Xue et al., 2022) have gained impressive improvement over ImageNet pre-training on various vision benchmarks. Very recent vision-language co-training approaches (Radford et al., 2021; Wang et al., 2022) demonstrate their extraordinary potential in the domain of multi-modal learning and applications. Yet, these generic representation learning methods adopt various data augmentation techniques to achieve translation and view invariance, while visuomotor driving sets in a highly dynamic environment. In this work, we show that the ever-victorious pre-training methods may not be the optimal choice, and introduce a curated paradigm for visuomotor driving policy learning.

**Pre-training for Visuomotor Applications.** Learning a control policy directly from raw visual input is challenging since the model needs to reason about visual pixels and dynamic behaviors simultaneously. Moreover, training visuomotor models from scratch usually requires tons of labeled data or environment interactions. To this end, recently, Shah & Kumar (2021) shows that feature representations from ResNet (He et al., 2016) pre-trained on ImageNet classification is helpful for RL-based dexterous manipulation tasks. Parisi et al. (2022) conducts extensive experiments on applying "off-the-shelf" pre-trained vision models in diverse control domains and validates their benefits to train control policies. CLIP (Radford et al., 2021) is also adopted in some embodied AI and robot navigation problems (Shah et al., 2022). Besides borrowing pre-trained weights for visuomotor tasks, researchers in robotics now desire a paradigm learning policy representations from raw data directly. Xiao et al. (2022); Radosavovic et al. (2022); Seo et al. (2022); Gupta et al. (2022) inherit the MIM spirit to realize visual pre-training for control tasks. Yang & Nachum (2021) investigates unsupervised representation learning objectives from D4RL environments (Fu et al., 2020), and Yamada et al. (2022) further adopts task-induced approaches to learn from prior tasks. However, compared with visuomotor driving, the visual inputs of such control tasks are less diverse which usually concentrate on objects and are much more compact.

To our best knowledge, ACO (Zhang et al., 2022b) is the only pre-training method customized for autonomous driving. By first training an inverse dynamic model on nuScenes (Caesar et al., 2020), they get pseudo steer labels of the driving videos and then construct the steer-conditioned discrimination for contrastive learning following MoCo. However, ACO ignores other crucial driving factors such as throttle and brakes, and its performance is largely limited by the inverse dynamic model. SelfD (Zhang et al., 2022a) is not strictly designed for pre-training while it also makes use of vast amounts of videos to learn driving policies via semi-supervised learning. It acquires the pseudo labeling knowledge from the target domain. These two methods both depend on the accuracy of pseudo labeling. In contrast, we realize fully self-supervised learning through dense geometric reconstruction, evading the possible adverse effect.

**Policy Learning for Autonomous Driving.** Visuomotor autonomous driving learns a driving policy directly from sensor inputs in an end-to-end manner (Codevilla et al., 2018; 2019; Liang et al., 2018; Chen et al., 2020a; Prakash et al., 2021; Chen et al., 2021; Wu et al., 2022; Shao et al., 2022; Hu et al., 2022; 2023). In essence, the inherent difficulty of the urban-style autonomous driving tasks makes such methods data-hungry. Interfuser (Shao et al., 2022), the current top-1 method on the CARLA Leaderboard (CARLA, 2022), requires 3 million labeled data samples for imitation learning (behavior cloning specifically). RL-based model MaRLn (Toromanoff et al., 2020) needs 20 million environment steps of interaction. The sample efficiency problem greatly impedes the real-world application of such approaches. In this work, we propose a self-supervised pre-training pipeline to learn driving policy related representations on unlabeled driving videos, and pave the way for these visuomotor autonomous driving models to further achieve satisfying performance.

## 5 CONCLUSION

In this work, we have proposed a fully self-supervised visuomotor driving policy pre-training paradigm PPGeo by modeling the 3D geometry of large-scale unlabeled driving videos. Taking a direct approach to infer the ego-motion and benefiting from the two-stage pre-training pipeline, we enable the visual encoder to learn driving policies based on single visual input. Our method outperforms the peer pre-training approaches by a large margin on a series of visuomotor driving tasks.

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
