# OpenReview forum: "Policy Pre-training for Autonomous Driving via Self-supervised Geometric Modeling"
_ICLR.cc/2023/Conference — ICLR 2023 poster_

### Official Review · Reviewer_hNsW · 2022-10-23

**Confidence:** 4
**Correctness:** 3
**Technical Novelty And Significance:** 2
**Empirical Novelty And Significance:** 3
**Recommendation:** 6

**Clarity, Quality, Novelty And Reproducibility:**

### Quality

The general quality of the work is sufficient.

### Novelty

The method proposed is novel in terms of the whole 3 steps framework that was built.
The individual steps can be found on previous works and the dataset used is also from a previous
work from the literature.

### Clarity and Reproducibility

One of the main drawbacks of this paper is exactly related to clarity and reproducibility. The paper lacks a clearer explanation of the method and implementation details which hinders its reproducibility. Details for that can be found on points one and two from the weaknesses section.

**Strength And Weaknesses:**

Strengths

1. The paper proposes a novel and general pre-training method that can be used on different contexts
and domains. This includes end-to-end driving in CARLA and odometry prediction on KITTI. It is capable of doing
so by training the backbone network to predict the whole photometric transformation and that correlates
more with actions than previous methods used in literature. The presented results have superior performance
than ImageNet pre-training, largely used on this domain.

2. The experimental section is very vast and clear. The experiments done in CARLA are complemented by
nu-scenes and KITTI.

3. Pre-trainings for the autonomous driving context are useful since the amount of data available is vast
and methods like this helps to improve diverse applications in autonomous driving which can benefit from
more effective pre-trainings than ImageNet.

Weaknesses

1.  I had several issues understanding exactly what was being proposed on this paper. The technical session is constituted by around only a single page with no supplementary material for a 3-steps algorithm. I think being concise on explanations is key but I felt I missed some details during this explanation. I was able to get the general idea easily but I still don't think I am able to get all the details even after reading the methodological session a few times.
     + The lack of a clear mathematical formulation for the whole system really hindered my understanding. For example how precisely, the phase one is used for phase 2 and then adapted for phase 3 (downstream tasks).
     + As quoted on page for "Then, in the second stage, we replace the PoseNet for ego-motion estimation with the visual encoder prepared for downstream driving policy learning tasks". This description in my opinion is kind of vague, and generate some doubts. What is does " prepared for downstream driving" means exactly ? This is replacing which parts of the network ?
    +  I understood that there are issues if we train the second phase directly but those were not clearly described. Why the second phase is so benefited by the first phase as it was shown on the ablation studies ? Could you provide more details on that ?
    + I missed some extra justification on using the photometric loss function. Isn't predicting pixels harder than pseudo-actions ? Maybe the paper would benefit going a bit deeper on this discussion. Maybe the fact that there is 2 phase pre-training helps the photometric prediction and makes this method more beneficial.


2. Following a bit the weakness 1 , the paper has, in my opinion, some reproducibility problems.
    +  I couldn't find the description of the architectures used for the different networks. I know that the policy network uses CIRLS architecture but how does that fit exactly with the other steps and the pre-training networks. I guess you have to be pre-training only the resnet-34 for all stages, but that is not clearly specified.
   + From the  metadriverse datatase that was was used.  How the 0.8 million images were chosen ? I feel the dataset has a big impact on performance for this case and there was little discussion on this.

3.  I missed comparison with papers that use action labels like [1] which can also be obtained freely on many of the domains the authors analysed.



[1] Xiao, Yi, et al. "Action-Based Representation Learning for Autonomous Driving." Conference on Robot Learning. PMLR, 2021.

**Summary Of The Paper:**

The paper proposes a two state pre-training technique that uses
unlabelled, uncalibrated driving data. This pre-training is used to improve
performance of policy training networks.

The two state consists of first, predicting the depth and pose
between consecutive frames by doing odometry and pose change estimation.
Then, as a second step, predict the motion based on a single frame directly.

The paper shows that those proxy tasks applied
sequentially can be used as a pre-training  that benefits CARLA imitation and reinforcement
learning policies and can also be used for odometry prediction and nu-scenes offline motion prediction.

**Summary Of The Review:**

The paper provides a useful and novel method for pre-training backbone networks
to be used on different autonomous driving tasks.
I would recommend clear acceptance, however I think there are reproduction and clarity issues that
still need to be addressed. With this I am recommending marginal acceptance.

---

> ### Author Response · Authors · 2022-11-16
> **Author response to Reviewer hNsW**
>
> Dear Reviewer hNsW,
>
> Thank you for your comments and suggestions. We address your concerns and questions below.
>
> >***Q1: Clear mathematical formulation for the whole system.***
>
> **A1:** For phase one, the PoseNet takes two consecutive images (i.e., $I_{t}, I_{t+1}$) as input, and the bottleneck feature goes to two MLP heads, one outputs 6-DoF ego-motion including translation and rotation between $t$ and $t+1$ and the other one outputs the intrinsic matrix including focal lengths and optical centers. The DepthNet takes $I_t$ as input and predicts a dense depth map. Then according to Equation 1 in the paper, we find correspondence between pixels in these two images based on the predictions (ego-motion, intrinsic matrix, and depth map). In the second phase, weights of PoseNet and DepthNet are frozen. We still use PoseNet to predict the intrinsic matrix and DepthNet to predict the depth map, but for ego-motion, we use another visual encoder (Vision Backbone + MLP head) instead. Now the visual encoder only takes $I_t$ as input, and it has to infer the future motion based on current visual input. In phase three (downstream tasks), the visual backbone in the pre-trained visual encode is used as the initial weight for downstream task models. We have added details about each model in the Supplementary.
>
> >***Q2: What does "prepared for downstream driving" means exactly.***
>
> **A2:** This means that the vision backbone (ResNet-34 in our case) of the visual encoder will be the same for downstream task models, so the pre-trained vision backbone could be used as an initial weight for downstream tasks.
>
> >***Q3: Why the second phase is so benefited by the first phase as it was shown on the ablation studies.***
>
> **A3:** The phase one is necessary since it provides a well-trained depth estimation network and an intrinsic estimation network. Predicting future ego-motion based on current input only is hard without direct supervision. And bad ego-motion prediction severely affects the learning process of the DepthNet. For example, if the motion predicted is too small compared with GT, to account for the view changes in the images, the DepthNet predicts very small depth around four corners of the image where the view change is relatively large. Therefore, directly training the visual encoder with DepthNet from scratch simultaneously degrades the quality for both motion and depth estimations.
>
> >***Q4: Extra justification on using the photometric loss function.***
>
> **A4:** Note that we are not directly regressing the pixel values (i.e., RGB values). With the estimated ego-motion, intrinsic matrix, and depth map, we build correspondence between pixels in $I_t$ and $I_{t+1}$ using Equation 1. Say pixel $p_t$ in $I_t$ corresponds to  $p_{t+1}$ in $I_{t+1}$, then the photometric loss is the $L_1$ difference between the RGB values of $p_t$ and $p_{t+1}$. As discussed in Section 2.2, the advantage of photometric loss over pseudo-actions is that the photometric reprojection loss enables each pixel to provide supervision so that inaccurate depth estimation in some pixels could be mitigated by accurate ones. But the ego-motion predicted from the PoseNet is a single scalar, which is sparse compared with the supervision from all pixels. The inferior quality of the pseudo-labels would greatly affect the learning of the visual encoder.
>
> >***Q5: Description of the architectures used for the different networks.***
>
> **A5:** Yes, we use ResNet-34 as the vision backbone and pre-train it in the visual encoder. We have added details about the architectures used for the different networks in the Supplementary. We will also make code and models publicly available.
>
> >***Q6: How the 0.8 million images were chosen.***
>
> **A6:** We just use the video data shared by ACO [1] in its repo. The video data is sampled at 1Hz following ACO [1] to obtain 0.8 million images in total. We agree with the reviewer that the dataset could make some influence which is a common problem in the computer vision field, and we think it is worth exploring in the future.
> >***Q7: Comparison with papers that use action labels like [2].***
>
> **A7:** Methods using action labels like [2] are direct imitation learning methods, which belong to supervised learning. The target of [2] is to predict affordances, and they train the model to predict control action with direct supervision only as a pre-training stage. In our case, the real-world video data has no action labels, and we target at proposing a fully self-supervised training scheme to utilize driving video without any label. Our pre-trained model can then be employed for tasks like imitation learning based tasks on dataset with action labels.
>
> > [1] Qihang Zhang et al. Learning to drive by watching youtube videos: Action-conditioned contrastive policy pretraining. ECCV, 2022.
> >
> > [2] Xiao, Yi, et al. "Action-Based Representation Learning for Autonomous Driving." Conference on Robot Learning. PMLR, 2021.

---

### Official Review · Reviewer_nuYx · 2022-10-25

**Confidence:** 4
**Correctness:** 3
**Technical Novelty And Significance:** 2
**Empirical Novelty And Significance:** 3
**Recommendation:** 5

**Clarity, Quality, Novelty And Reproducibility:**

**Clarity, Quality:** The high-level ideas are clear, but the figures and results section needs more polish.
**Novelty:** I found the proposed method to be relatively novel. Even though the heavy lifting of their pre-training stage builds on established methods, showing this idea actually works in end-to-end planning was novel.
**Reproducibility:** The authors provide thorough details about their method and experiments, I am fairly confident this could be reproduced.

**Strength And Weaknesses:**


## Strengths

* Good high-level goal - using large amounts of unlabeled data as pre-training
* Variety of experiments on real world datasets (nuScenes) in addition to the CARLA simulator
* Clear writing + notation, particularly method and description of baselines

## Weaknesses

* Table 1,2, 3, 4 show competitive reslts against other pre-training methods, but are missing **all** baselines from literature.
* Visual encoder seems redundant with PoseNet. In Sec 2.2, rather than using the PoseNet which operates on consecutive frames, the authors use a new visual encoder on single camera images since the authors claim "single input setting aligns with downstream driving tasks". I would argue the contrary that driving should be done using more than one image. Additionally, operating on image sequences allow for the pre-trained PoseNet to directly act as the visual encoder, simplifying Stage 2 of the pipeline.
* Section 2.2 "ego-motion predicted from the PoseNet is too spare": This claim goes without evidence.

## Minor Issues

* Figure 1: Hard to see how this directly motivates the work.
* Figure 2: I am confused why "... Since a car is ahead, We need to STOP" is there.
* Page 7 is quite busy

**Summary Of The Paper:**

This work targets the end-to-end autonomous driving task from monocular images.
In particular, they propose **PPGeo**, and leverage large-scale unlabeled driving videos mined from the web in order to pre-train a visual encoder.
To do this, they first train a DepthNet and PoseNet on their unlabeled dataset, using consecutive images $I_i, I_{i+1}$ to predicting camera intrinsics $K$, depth $D$, and ego-motion $T$ to minimize a photometric loss, following Monodepth2.
In the second stage of pre-training, they freeze the PoseNet and DepthNet, train a separate visual encoder to predict ego-motion from a *single image* and minimize the same loss.
By learning ego-motion, this visual encoder learns important features which are directly aligned with the downstream task (planning).
They show improved results on the CARLA simulator and real-world nuScenes dataset (open-loop).

**Summary Of The Review:**

Leveraging large amounts of unlabeled data as a self-supervised pre-training tasks for autonomous driving is a great unexplored direction and well motivated. The writing is clear and I am glad to see experiments across a variety of benchmarks. However, there are a few issues that I currently have that leans me against acceptance - (1) we need to see how their method performs against the current SOTA, not just the self-supervised baselines and (2) question some parts of the design of the pipeline (throwing away the ego-motion from the PoseNet after stage 1).

---

> ### Author Response · Authors · 2022-11-16
> **Author response to Reviewer nuYx**
>
> Dear Reviewer nuYx,
>
> Thank you for your helpful review. We address your concerns below.
>
> >***Q1: Only results against other pre-training methods, but are missing all baselines from literature.***
>
> **A1:** In this work, we design a pre-training method for end-to-end autonomous driving tasks, and do not aim at proposing a certain new model or architecture for autonomous driving. It is reasonable and fair to compare our method with other pre-training methods on autonomous models. For the Navigation and Navigation Dynamic task, we compare pre-training methods on the classic model CILRS [1]. And for the Leaderboard Town05-long task, we use the **SOTA** end-to-end autonomous driving method TCP [4] for comparison. In the rebuttal, we provide another experiment (AIM + PPGeo), and comparisons to end-to-end autonomous driving baselines on this task in the table below. It is added in the Supplementary in the revised version as well.
>
> |     Methods     |   Driving Score  | Route Completion |
> |:---------------:|:----------------:|:----------------:|
> |       LBC [1]       |    7.05 ± 2.13   |   32.09 ± 7.40   |
> |       AIM [2]      |   26.50 ± 4.82   |   60.66 ± 7.66   |
> |    Transfuser [2]   |   33.15 ± 4.04   |   56.36 ± 7.14   |
> |       NEAT [3]     |   37.72 ± 3.55   |   62.13 ± 4.66   |
> |       TCP [4]      |   41.29 ± 3.20   |   57.52 ± 4.87   |
> | **AIM + PPGeo** | **33.94(+7.44) ± 5.03** | **59.09 ± 5.62** |
> | **TCP + PPGeo** | **47.44(+6.15) ± 5.63** | **65.05 ± 5.11** |
>
> >***Q2: Visual encoder seems redundant with PoseNet. Driving should be done using more than one image.***
>
> **A2:** It is necessary to have both the PoseNet and a visual encoder. The PoseNet in the first stage takes the current image and the image in the **future step** as input to infer the pose change. However, in autonomous driving, it is **impossible to have future images** and we need to make driving decisions only on current and past inputs. Therefore, in the second stage, the visual encoder has to infer the future motion based on the current image only, enabling it to reason about the driving decision based on the current image. The claim "single input setting aligns with downstream driving tasks" probably causes some misunderstandings. We want to emphasize that the downstream driving tasks can not access future images. Yes, it is possible to use multiple inputs including **the current one and past ones** for driving tasks. Here we choose to use the current input only for simplicity. We have modified the claim to avoid possible ambiguity.
>
> >***Q3: The claim "ego-motion predicted from the PoseNet is too spare" goes without evidence.***
>
> **A3:** Immediately after the claim "ego-motion predicted from the PoseNet is too spare" in Section 2.2, we explain that in our approach the photometric reprojection error enables each pixel to provide supervision so that inaccurate depth estimation in some pixels could be mitigated by the accurate ones. But the ego-motion predicted from the PoseNet is a single scalar, which is sparse compared with the supervision from all pixels.
>
> >***Q4: How Figure 1 directly motivates the work.***
>
> **A4:** In Figure 1, we emphasize the uniqueness of visuomotor driving policy learning. Visuomotor driving policy learning requires the model to **focus on very specific information** (e.g., traffic signal) in the visual input and ignore other redundant ones (e.g., static street lights and background buildings, light and weather conditions). This makes this task different from general detection and segmentation tasks where the model needs to focus on all the interested objects or semantic categories over the whole input. This motivates us to design a specific pre-training paradigm.
>
> >***Q5: In Figure 2, why "... Since a car is ahead, We need to STOP" is there.***
>
> **A5:** In this example, the future motion of the ego vehicle is to stay still (stop). The PoseNet can make this prediction based on the fact that the current image and the future image are almost the same (Stage One). As for the visual encoder, it has to infer the motion from the current image only (Stage Two). Therefore, it reasons about the true cause of this stop motion or brake decision, that is "a car is ahead".
>
> >***Q6: Page 7 is quite busy.***
>
> **A6:** Thank you for your suggestion. We have adjusted it accordingly in the updated version.
>
> > [1] Dian Chen et al. Learning by cheating. CoRL, 2020.
> >
> > [2] Aditya Prakash et al.  Multi-modal fusion transformer for end-to-end autonomous driving. CVPR, 2021.
> >
> > [3] Chitta, Kashyap et al. NEAT: Neural Attention Fields for End-to-End Autonomous Driving. ICCV, 2021.
> >
> > [4] Penghao Wu et al. Trajectory-guided Control Prediction for End-to-end Autonomous Driving: A Simple yet Strong Baseline. NeurIPS, 2022.

---

### Official Review · Reviewer_Wqac · 2022-10-25

**Confidence:** 4
**Correctness:** 4
**Technical Novelty And Significance:** 3
**Empirical Novelty And Significance:** 3
**Recommendation:** 8

**Clarity, Quality, Novelty And Reproducibility:**

It will be good if the authors do a better job in describing what policy decisions in autonomous driving conditions are. Though some examples are shown in Figure 1 and Figure 2, it will be better if these policies are explained in more detail with more examples.

Lots of autonomous driving tasks are described in Section 3.2. It will help if these are explained better with examples (possibly visual illustrations wherever possible).

It is not clear what metric is shown from Table 1 and Table 2. Many metrics are mentioned in Table 3,4,5, without any reference in the body of the paper.

The activation maps from Figure 4 do not provide much insight. Almost all the maps of the proposed method are in the center. It will help if more diversity is presented.

It will be good to discuss some failure cases (or some case studies) of the proposed method. Since this paper is in the autonomous driving domain, knowing about failures is important.


**Strength And Weaknesses:**

The proposed method will be an important contribution in the autonomous driving field. The use of self-supervised learning on large public datasets and utilizing them to learn driving policy is certainly useful. The usefulness of this general method to depth and odometry estimation is also a plus.

Though the paper’s contribution is important, there are some issues in the presentation of the paper, which if fixed, will make this paper even better.


**Summary Of The Paper:**

This paper proposes PPGeo - Policy Pre-training via Geometric modeling,  a driving policy paradigm which uses a self-supervised framework for policy pretraining in visuomotor driving. Policy representations are learnt by modeling 3D geometric scenes (pose and depth) on public datasets. This is in turn done in two stages. In the first stage, the method generates pose and depth predictions with two input consecutive frames. In the second stage, driving policy representations are learnt from a single image by predicting the future ego motion. Experiments are conducted on lots of datasets under different conditions and applications.


**Summary Of The Review:**

The proposed method in the paper appears to be of good value. With improved presentation, the paper can get better rating.

---

> ### Author Response · Authors · 2022-11-16
> **Author response to Reviewer Wqac**
>
> Dear Reviewer Wqac,
>
> Thank you for appreciating our work. We address your concerns below.
>
> >***Q1: What policy decisions in autonomous driving conditions are.***
>
> **A1:** For end-to-end (visuomotor) autonomous driving, the driving policy is conditioned on the visual input (i.e., the front-view image) and other possible measurements like ego speed and navigation information. We will add more explanations and examples to better illustrate the idea.
>
> >***Q2: Autonomous driving tasks in Section 3.2 should be explained better with examples.***
>
> **A2:** Thank you for the suggestion. We have added more descriptions and visual examples for each task in the Supplementary.
>
> >***Q3: Metrics not clear or not referenced in the body of the paper.***
>
> **A3:** Thanks for the reminder. We have described the metrics for each task in the revised version (marked as blue texts).
>
> >***Q4: Activation maps from Figure 4 do not provide much insight.***
>
> **A4:** For autonomous driving, most of the time we should focus on the center to capture the directed related road/vehicle/traffic lights information. In the activation map examples, our model focus on the road in front of the vehicle in cases in Row 1&2; our model captures the stopped front vehicle in cases in Row 3&4; and the traffic light is captured by our method in Row 5.
>
> >***Q5: Failure cases of the proposed method.***
>
> **A5:** Yes, we agree that failure case studies are very important for autonomous driving. We have added corresponding discussions in the Supplementary.

---

### Official Review · Reviewer_DZe7 · 2022-10-29

**Confidence:** 2
**Correctness:** 3
**Technical Novelty And Significance:** 1
**Empirical Novelty And Significance:** 3
**Recommendation:** 6

**Clarity, Quality, Novelty And Reproducibility:**

As the authors point out, there are plenty of other papers that also seek to pre-train self-driving models on unlabeled YouTube videos. The authors should correct me if I'm wrong, but my sense is the main novelty in this paper is that they are using the photometric error to train their future-pose prediction model, instead of using a pre-processing stage to estimate future-pose prediction then train the model to predict those labels. Beyond that, I think the experiments are high-quality and the method and experiments were presented clearly.

**Strength And Weaknesses:**

This paper presents an interesting and convincing application of unsupervised depth models for pre-training end-to-end self-driving policies. I found the empirical analysis very thorough; the authors include open-loop and closed-loop experiments in CARLA as well as an open-loop real-world task on nuScenes. I also like the paper at a conceptual level. Unsupervised depth models certainly learn valuable representations and I think it makes sense to probe the extent to which these representations can enable other tasks.

The main weakness that I see is primarily with respect to impact. I'm wondering if the authors have tested how well their model can be used as a pre-training stage for maybe object detection or segmentation? In terms of applicability to self-driving, demonstrating that the author's method boosts detection performance would be a huge win. In many ways, I think the results for motion prediction are very intuitive in that the pre-training stage is basically imitation learning on 1-second long future trajectories, so the main difference between the pre-training stage and testing stage is domain gap. To claim this method is useful for pre-training, I think it's important to show task generalization.

I have a few other comments and edits below:

- page 3 contribution 1 - authors should clarify the scope of this claim. I don't think the goal of this paper is significantly different enough from [1] to be able to claim their paper is the first to attempt this task. If the "without pseudo-labels" part is important, the authors should define pseudo-labels and explain why it's significant to avoid using them.

- page 4 - "casual" should be "causal" I think?

- I recommend visualizing Table 1 as a graph of performance vs. # training samples, with a curve for each pre-training method

- Table 3 and Table 5 - bold the numbers that are better to improve interpretability

- SelfD baseline - SelfD seems more general than the authors' approach in that it trains the encoder to predict the future pose for multiple future timesteps. Could the authors comment on why the authors choose to train their visual encoder to predict only the pose 1 second into the future?

- Have the authors ablated the choice of sampling the videos at 1 Hz for pre-training? I'm wondering if it's important that the pre-training frequency roughly matches the CARLA frequency. If they match, is it correct to say that the pre-training task is equivalent to imitation learning? My sense though is that ACO and SelfD also pre-train with different variants of imitation learning.

[1] "Learning to drive by watching youtube videos" Qihang Zhang, Zhenghao Peng, and Bolei Zhou.

**Summary Of The Paper:**

This paper proposes a new pre-training task tailored for end-to-end self-driving. The idea is to take an unsupervised monocular depth model (for instance [1]) and distill the component that takes as input 2 images and predicts the pose between them into a single-image model. The resultant model therefore learns a representation that is useful for predicting short-term future motion of the ego vehicle. The authors get good results when using this network as an initialization for 3 imitation learning CARLA tasks, 1 RL CARLA task, and 1 imitation learning nuScenes task.

[1] "Digging Into Self-Supervised Monocular Depth Estimation"

**Summary Of The Review:**

I think this paper is a principled study in how one can leverage unsupervised depth prediction for pre-training driving policies. Ideally, the authors would demonstrate to what extent the model can be used for standard self-driving perception tasks such as object detection. For the current tasks, it seems mostly like a domain adaptation paper in which the authors train for motion prediction on YouTube videos then fine-tune on synthetic or nuScenes images, in which case I think it's quite intuitive that this pre-training strategy would outperform for instance image classification (Imagenet). If the authors can expand the set of tasks for which their method is a top-performing pre-training strategy, the impact of the paper will increase substantially.

---

> ### Author Response · Authors · 2022-11-16
> **Author response to Reviewer DZe7 (2/2)**
>
>
> >***Q3: Comparison with the SelfD baseline. Why predict only the pose 1 second into the future.***
>
> **A3:** SelfD [1] is not strictly a pre-training method. Its predicted representation is based on the downstream task, and it has to go through the *training on the task data -> training on the YouTube data with pseudo-label -> fine-tuning on the task data* pipeline for **every single task**. On the contrary, pre-training should only train the model once for all tasks. And the performance of SelfD would be severely affected if the size of the task data is limited, which makes it actually not general. The reason for only predicting 1 second into the future in our method includes two aspects:
>
> 1) Our goal is to let the visual encoder learn the general "driving decision/intention" (e.g., turn left, turn right, stop). The motion in 1 second is generally enough to contain such information. Our goal is not to fully predict a planned trajectory in the pre-training stage or directly imitate the downstream tasks, and downstream tasks may also predict direct control action instead of trajectory predictions.
>
> 2) Predicting a longer time or multiple points causes bigger changes to the front-view images, leading to more occlusions and larger motions from other dynamic objects. This would harm the photometric reprojection based self-supervised learning process.
>
>
> >***Q4: Choice of sampling rate. If the pre-training task is equivalent to imitation learning.***
>
> **A4:** We follow ACO [2] to sample the video at 1Hz to obtain the same amount of total training data for a fair comparison. We experiment to use half of the data with a sample frequency 2Hz to keep the same total number of training data. We test it on the Closed-loop Navigation task, and the results (Success Rate) are shown below. The performance of 2Hz decreases, probably due to less diverse data. Interestingly, the training data for CARLA related tasks is collected at 2Hz. Therefore, we don't observe a clear relationship between frequency matching and imitation learning. Our pre-training tasks could be viewed as imitation learning to some extent as we are imitating the driving intentions contained in the driving video. However, we are **not directly imitating** the downstream tasks, and the data frequency for downstream tasks and pre-training are not directly related.
>
> | Sampling Frequency |   4K data   |   8K data   |   16K data  |   40K data  |
> |:------------------:|:-----------:|:-----------:|:-----------:|:-----------:|
> |         1Hz        | 42.0 ± 2.0 | 73.3 ± 6.1 | 91.3 ± 1.2 | 96.7 ± 1.2 |
> |         2Hz        | 30.0 ± 2.0 | 58.0 ± 3.5 | 89.3 ± 6.1 | 94.0 ± 2.0 |
>
>
>
> >***Q5: The main novelty is using photometric error to train the future-pose prediction model.***
>
> **A5:** Yes, our main novelty is designing a scheme to let the visual encoder learn the driving intentions/decision in the unlabeled data in a fully self-supervised way, and this is implemented using the photometric error.
>
> >***Q6: Minor issues about writing and tabels.***
>
> **A6:** Thank you for your valuable suggestions. Yes, 'casual' should be 'causal' in page 4. We have corrected it and modified tabels as suggested in the revised version.
>
>
> > [1] Jimuyang Zhang et al. Selfd: Self-learning large-scale driving policies from the web. CVPR, 2022.
> >
> > [2] Qihang Zhang et al. Learning to drive by watching youtube videos: Action-conditioned contrastive policy pretraining. ECCV, 2022.

---

> ### Author Response · Authors · 2022-11-16
> **Author response to Reviewer DZe7 (1/2)**
>
> Dear Reviewer DZe7,
>
> Thank you for your thoughtful comments and suggestions. We address your concerns on the weaknesses below.
>
> >***Q1: Possible impact on other tasks like object detection or segmentation. The main difference between the pre-training stage and testing stage is domain gap.***
>
> **A1:** We agree with the reviewer that applying our pre-trained knowledge on more tasks could be impactful. We address this problem in two aspects:
>
> 1) The purpose of our policy pre-training is to let the visual encoder learn to focus on **certain information critical for driving policy learning**, which is useful for downstream end-to-end driving tasks. Note that our testing stage (downstream tasks's domain) is not just trajectory prediction, but general driving policy learning such as control action predictions, trajectory planning, etc. The pre-trained visual encoder is expected to focus on specific regions only (e.g., the front car and traffic light) while ignoring other redundant information irrelevant to driving decisions (e.g., street lights and buildings on the sides). This is different from other autonomous driving tasks like detection/segmentation which require the model to focus on all the interested objects or semantic categories over the whole input.
>
> 2) Nevertheless, among the autonomous driving tasks, the lane detection task may also benefit from our policy pre-training, since lanes are crucial for predicting driving decision/future trajectory. Therefore, we compare ImageNet pre-trained backbone and ours on the CULane 2D lane detection task. We use 10% training data to train several popular 2D lane detection methods. The F1-score is shown in the table below, and we can observe our method brings performance gain to the 2D lane detection task for all detectors. Besides, as shown in Section 3.4, the DepthNet and PoseNet in our pre-trained framework could also bring performance improvement to depth and odometry estimation tasks. Additionally, inspired by previous work [1] which proves that depth estimation as the pre-training task could help 3D detection tasks, we will further explore applying the pre-trained DepthNet to monocular 3D detection models.
>
> |  | LaneATT [2] |  RESA [3]  |  UFLD [4] | SCNN [5]  |
> |:--------:|:---------:|:------:|:------:|:--------:|
> | ImageNet | 65.90  | 57.85 | 59.68 | 46.03 |
> | **PPGeo** (Ours) | **67.63(+1.73)**  | **60.14(+2.29)** | **60.72(+1.04)** | **47.53(+1.50)** |
>
> >***Q2: Scope of the claim in contribution 1. Definition of pseudo-labels.***
>
> **A2:** Yes, "without pseudo-labels" is important here. In contribution 1, we claim that we are "the first attempt to achieve a fully self-supervised framework without any need of pseudo-labels". "Pseudo-labels" here means using labels in the pre-training process from an additional model trained on an extra dataset with annotations. As discussed in Intro and Section 4.1(Pre-training for Visuomotor Applications), since pseudo-labels usually come from predictions of another model, the quality of the pre-training is severely restricted by the predicted pseudo-labels and the extra datasets. Meanwhile, using extra datasets with annotations makes it not fully self-supervised/unsupervised. We have added definitions and explanations (marked as blue texts) in the revised paper to better clarify. Thank you for pointing this out.
>
> > [1] Park Dennis, et al. Is pseudo-lidar needed for monocular 3d object detection. ICCV, 2021.
> >
> > [2] Lucas Tabelini, et al. Keep your Eyes on the Lane: Real-time Attention-guided Lane Detection. CVPR, 2021.
> >
> > [3] Tu Zheng, et, al. RESA: Recurrent Feature-Shift Aggregator for Lane Detection. AAAI, 2021.
> >
> > [4] Zequn Qin, et, al. Ultra Fast Structure-aware Deep Lane Detection. ECCV, 2020.
> >
> > [5] Xingang Pan, et, al. Spatial As Deep: Spatial CNN for Traffic Scene Understanding. AAAI, 2018.

---

### Author Response · Authors · 2022-11-16
**[Update] General Author Response for Rebuttal**

Update:

Hi Reviewers / ACs,
The author-reviewer discussion will be closed soon. Please let us know if you have further questions or concerns. We look forward to your feedback and discussion.

Best,
Authors

------
Dear Area Chairs and Reviewers,

We appreciate all the reviewers for their careful reviews and valuable comments. Overall, all reviewers recognized the motivation and novelty of our work and acknowledged the extensive experiments and good performance.

However, there exists some ambiguity about the specific process of our proposed method, which causes some concerns about the reasonability of certain parts (nuYx, hNsW). Besides, there are some valuable suggestions about the presentation and writing of our work (DZe7, nuYx).

Next, we provide detailed responses to the concerns of all reviewers and revise our paper based on the suggestions to include more clarification and details to eliminate possible misunderstandings and ambiguity. Please see each response below and the revised paper.

Thanks.

---

### Decision · Program_Chairs · 2023-01-20

**Decision:**

Accept: poster

**Justification For Why Not Higher Score:**

However, all the reviewers noted that (1) the paper is not well polished (which may be fixed in camera ready though), and (2) the method is ok for an acceptance but not exciting. So we would like to recommend an accept (poster) for this paper.

**Justification For Why Not Lower Score:**

The AC and reviewers agreed on the strengths of this paper summarized above and reached a consensus that the paper could be a solid contribution to the community and worth it for appearing at ICLR.

**Metareview: Summary, Strengths And Weaknesses:**

This paper proposes a new pre-training method tailored for end-to-end autonomous driving from monocular images, PPGeo, which learns policy representations from large-scale unlabeled and uncalibrated YouTube driving videos in a two-stage fashion. The method is novel and useful and has the potential to be applied to different contexts and domains. The empirical analysis and results are very thorough and validate the effectiveness of the proposed approach, and self-supervised pre-training is indeed a promising direction and necessary step for the research field of autonomous driving. On the other hand, reviewers also noted that the paper is not well polished and the method is worth it for appearing at ICLR but not very exciting. Overall, the AC and the reviewers are positive about the contributions of this work and would like to see it in the conference if space allows.

**Note From Pc:**

if the above contains the word "oral" or "spotlight" please see: "oral" presentation means -> notable-top-5% and "spotlight" means -> notable-top-25%. As stated in our emails, we are disassociating presentation type from AC recommendations

**Summary Of Ac-Reviewer Meeting:**

The AC and reviewers agreed on the strengths of this paper summarized above and reached a consensus that the paper could be a solid contribution to the community and worth it for appearing at ICLR. Reviewer nuYx was concerned about lack of comparison with supervised methods, but during the discussion, Reviewer nuYx realized that the rebuttal had resolved this concern, so they were willing to raise the review score from 5 to 6. So the reviewing score should be 8 6 6 6 now.

However, all the reviewers noted that (1) the paper is not well polished (which may be fixed in camera ready though), and (2) the method is ok for an acceptance but not exciting. So we would like to recommend an accept (poster) for this paper.